

# Genetic variability and trait associations for physiological and agronomic characteristics in bread wheat genotypes under drought stress and well-watered conditions

Mohammed O. Alshaharni[1], Fatmah A. Safhi[2], Nora M. Al Aboud[3], Dmitry E. Kucher[4], Eman Fayad[5], Mohammed Alqurashi[5], Rahmah N. Al-Qthanin[1,6], Ibtesam S.M. Almami[7], Heba I. Ghamry[8], Diaa Abd El-Moneim[9], Mohamed M. Kamara[10] and Abdelraouf M. Ali[4,11]

[1] Biology Department, College of Science, King Khalid University, Abha, Saudi Arabia
[2] Department of Biology, College of Science, Princess Nourah bint Abdulrahman University, Riyadh, Saudi Arabia
[3] Department of Biology, Faculty of Science, Umm Al-Qura University, Makkah, Saudi Arabia
[4] Department of Environmental Management, Institute of Environmental Engineering, RUDN University, Moscow, Russia
[5] Department of Biotechnology, College of Sciences, Taif University, Taif, Saudi Arabia
[6] Prince Sultan Bin Abdelaziz for Environmental Research and Natural Resources Sustainability Center, King Khalid University, Abha, Saudi Arabia
[7] Department of Biology, College of Science, Qassim University, Buraydah, Saudi Arabia
[8] Nutrition and Food Science, Department of Biology, College of Science, King Khalid University, Abha, Saudi Arabia
[9] Department of Plant Production, (Genetic Branch), Faculty of Environmental and Agricultural Sciences, Arish University, El-Arish, Egypt
[10] Department of Agronomy, Faculty of Agriculture, Kafrelsheikh University, Kafr El-Sheikh, Egypt
[11] National Authority for Remote Sensing and Space Sciences (NARSS), Cairo, Egypt

Corresponding authors
Diaa Abd El-Moneim,
dabdelmoniem@aru.edu.eg
Abdelraouf M. Ali,
raouf.shoker@narss.sci.eg

## ABSTRACT

Drought is a critical abiotic stress significantly reducing global wheat production, especially under climate fluctuations. Investigating wheat genetic variability using physiological and agronomic characteristics is essential for advancing breeding to enhance drought resilience and ensure sustainable production in light of global population growth. The genetic diversity and associations among traits of fourteen diverse genotypes of bread wheat in drought-stressed and well-watered conditions were studied, focusing on physiological and agronomic responses. Significant variations were detected among irrigation regimes, genotypes, and their interactions for all assessed characteristics. Drought stress substantially declined chlorophyll $a$ (Chl $a$) and $b$ (Chl $b$), net photosynthetic rate (NPR), transpiration rate (Tr), stomatal conductance (gs), membrane stability index (MSI), relative water content (RWC), plant height (PH), yield-related attributes, and grain yield. Conversely, it significantly increased malondialdehyde content, proline content (ProC), and activities of antioxidant enzymes, including catalase (CAT), ascorbate peroxidase (APX) and superoxide dismutase (SOD). The genotypes, G3 (L-1117), G8 (L-120), and G12 (L-1142) exhibited superior drought tolerance, maintaining high photosynthetic efficiency, RWC, antioxidant

enzyme activity, and grain yield. Under drought conditions, these genotypes achieved grain yields of 6.32 t/ha (G8), 5.97 t/ha (G12), and 5.84 t/ha (G3), significantly surpassing the other genotypes. Genotypic classification and drought tolerance indices confirmed the superiority of G3, G8, and G12 as drought-resilient candidates, while G2, G5, G7, and G14 exhibited lower adaptability. Genotypic stability analysis (additive main effects and multiplicative interaction (AMMI) and ranking biplot) indicated that G3, G8, G6, and G12 were highly stable across diverse environments, making them promising candidates for wheat breeding programs. Agronomic traits such as PH, number of grains per spike (NGPS), and thousand kernel weight (TKW) were positively associated with drought tolerance. Furthermore, the multivariate analyses, including principal component analysis (PCA), correlation, and path analysis, highlighted the significance of RWC, MSI, chlorophyll content, and antioxidant enzymes in sustaining yield under drought stress. Broad-sense heritability estimates were high for key drought-related traits, particularly APX, SOD, and NGPS, indicating strong genetic potential for selection. These findings indicated the importance of integrating physiological and biochemical markers into breeding programs to develop high-yielding drought-tolerant wheat varieties, contributing to sustainable wheat production under water-limited conditions.

ltivariate analyses

# INTRODUCTION

Wheat (*Triticum aestivum* L) is one of the most commercially significant and widely cultivated crops in the world (*Mokhtari, Majidi & Mirlohi, 2024*). It is cultivated on 220.4 million hectares, producing approximately 798.9 million tons annually (*FAOSTAT. 2023*). It is a dietary staple in many countries, primarily used for bread and various baked products (*Mesta-Corral et al., 2024*). Furthermore, its straw is utilized for animal feed and in the manufacturing of diverse industrial products (*Kamara et al., 2021*). Wheat grains are distinguished by their high carbohydrate content, which is a vital energy source. Its adaptability and nutritional advantages are fundamental components of global diets, contributing to food security and economic resilience. Wheat production should be improved to meet the dietary needs of the growing global population (*Neupane et al., 2022*).

Climate fluctuations pose a significant challenge to global wheat production (*Rezaei et al., 2023*). Increasing temperatures and frequent variations in precipitation are expected to increase drought severity worldwide (*Bracho-Mujica et al., 2024*). The water deficit is a severe environmental challenge that significantly impacts wheat production (*Mao et al., 2023*). It induces biochemical, physiological, and morphological alterations in the plants, disrupting their growth and development (*Farooq et al., 2024*). The severity of drought stress and its detrimental effects on plants depend on the duration, intensity, and timing of water deficit in relation to the plant growth stage (*Wang et al., 2022*). Specifically, cell

dehydration caused by water scarcity restricts cell elongation, induces stomatal closure, reduces photosynthetic efficiency, and restricts overall plant growth and development (*McAusland et al., 2020*). The primary way that plants respond to water scarcity is through closing their stomata, which reduces water loss from plant leaves (*Qiao et al., 2024*). Moreover, reactive oxygen species (ROS) are produced by imbalanced photochemical reactions in chloroplasts, which result in surplus light energy that is not effectively used in photosynthesis. The oxidative stress damages cellular structures, including membranes, through lipid peroxidation (*Sachdev et al., 2021*). Drought-induced damage disrupts plant growth, impairs mineral uptake, and compromises photosynthetic activity. To mitigate oxidative stress caused by ROS, plants activate their antioxidant defense systems. Enzymes such as superoxide dismutase (SOD), catalase (CAT), and ascorbate peroxidase (APX) play a crucial role in neutralizing ROS and maintaining cellular integrity. SOD serves as the first line of defense by converting superoxide radicals ($O_2^-$) into hydrogen peroxide ($H_2O_2$) and oxygen ($O_2$). Subsequently, CAT and APX detoxify $H_2O_2$, preventing oxidative damage and ensuring cellular stability under stress conditions (*Gill & Tuteja, 2010*). Additionally, drought stress often leads to increased accumulation of proline, a key osmoprotectant (*Zia et al., 2021*). Proline stabilizes cell membranes, mitigates oxidative damage, and supports plant ability to tolerate water scarcity (*Shafi, Zahoor & Mushtaq, 2019*).

Wheat production in Egypt faces significant challenges due to the limited and unpredictable rainfall (*Tadesse, Bishaw & Assefa, 2019*). The reliance on irrigation, particularly through the Nile-based irrigation network, has been essential for sustaining wheat production (*Nikiel & Eltahir, 2021*). Yet, this reliance is increasingly strained by growing water scarcity and the impacts of climate change, which exacerbate issues of water availability and variability in precipitation patterns. The growing frequency of drought conditions, especially during critical growth stages like flowering and grain filling, has led to reduced wheat yields (*Mannan et al., 2022*). As a result, there is a pressing need to develop wheat varieties that are more resilient to drought stress (*Ezzat et al., 2024*).

Traditional wheat breeding programs often rely on univariate statistical approaches (*Ormoli et al., 2015*) which are useful but have limitations in addressing the complex interactions among multiple traits and environmental stress factors. Otherwise, advanced multivariate statistical techniques, such as principal component analysis, cluster analysis, and additive main effects and multiplicative interaction (AMMI) analysis allow for the identification of key traits, genotype performance assessment, and classification of genotypes based on multiple performance criteria (*Galal et al., 2023*). Understanding the interrelationships between grain yield and key physiological parameters is essential for enhancing breeding programs aimed at drought tolerance (*Abd-El-Aty et al., 2024*). Identifying traits that serve as reliable selection tools under both drought-stressed and well-watered conditions enables breeders to develop more targeted and efficient screening methods (*Mousa et al., 2024*). Physiological traits such as proline accumulation, antioxidant activity, relative water content (RWC), and chlorophyll concentration have been recognized as valuable secondary markers for identifying drought-tolerant genotypes (*Morsi et al., 2023*). Integrating these agronomic and physiological traits into breeding programs provides
a holistic approach to improving wheat resilience to drought stress while maintaining high grain yield potential (*Sakran et al., 2022*).

Despite extensive research on drought tolerance in wheat, there remains a significant gap in understanding the genetic variability and phenotypic associations of key physiological, biochemical, and agronomic traits under both drought-stressed and well-watered conditions. Hence, this research investigated the genetic diversity and trait associations of fourteen diverse bread wheat genotypes to understand their drought resilience. By identifying key traits such as chlorophyll content, RWC, membrane stability index (MSI), and antioxidant enzyme activities, this study could offer valuable information for breeders looking to develop drought-tolerant wheat varieties with high yield potential under water-limited environments.

## MATERIALS & METHODS

### Experimental site

Field experiment was conducted during two growing seasons of 2021–2022 and 2022–2023 at Kafr El-Sheikh governorate, located in the North Egypt (31°18′N, 30°46′E). For the 2021/22 season, the maximum temperatures ranged from approximately 25 °C in November to a peak of around 30 °C in March, while minimum temperatures varied between 15 °C and 20 °C. Rainfall during this season fluctuated significantly, with the highest amount recorded in March, reaching around 70 mm, while some months, such as January and February, experienced minimal rainfall. In the 2022/23 season, temperatures were similar, with maximum temperatures reaching up to 30 °C during March. However, this season had more rainfall in March with over 100 mm recorded (Fig. S1). The soil at the experimental site was characterized as clay (49% clay, 14.5% sand, and 36.5% silt) throughout the profile. In addition, the soil water parameters included a permanent wilting threshold of 20.3%, a field capacity of 37.2%, and an accessible water content of 16.4% (Table S1). Electrical conductivity (EC) was 2.35 dS/m and the organic matter content averaged 1.7%, and available phosphorus increased slightly to an average of 9.15 ppm. Total nitrogen averaged 589 ppm, and exchangeable potassium (K) was 444 ppm.

### Plant materials

Fourteen diverse bread wheat genotypes were selected based on their known variability in drought tolerance and yield potential, which were confirmed through preliminary screening. The genotypes represent a range of performance levels under both drought-stressed and well-watered conditions, making them good candidates for investigating genetic diversity and trait associations related to drought resilience. The study utilized diverse genotypes, including six advanced breeding lines that have undergone multiple selection cycles for desirable agronomic traits, five exotic genotypes, sourced from CIMMYT, were included to enhance genetic diversity. The genotypes also included three high-yielding commercial cultivars, that have been commercially released and recognized for their superior productivity under optimal conditions. Detailed information on the pedigrees and origins of these genotypes is provided in Table S2.

## Experimental design and agronomic practices

The study used two distinct irrigation treatments to evaluate the wheat genotypes under different water availability conditions. The first treatment was well-watered conditions, which applied 4,448 m$^3$/ha of water over five irrigations during the growing season. This ensured that the plants received adequate water to grow optimally. The drought stress treatment applied only 2,865 m$^3$/ha of water, spread over just two irrigations during the season, simulating drought conditions typical of drought-prone environments. The applied two irrigation regimes were strategically timed to impose stress during critical growth stages, such as flowering and grain filling, when water limitations are most detrimental to wheat yield. The timing of irrigation relative to wheat growth stages was a critical aspect of the drought stress treatment. The well-watered plants received irrigations throughout the season to maintain optimal water availability, while the drought-stressed plants experienced water withholding during the critical reproductive stages, intensifying the water stress impact on plant growth and yield. The drought treatment was applied at key developmental stages of the wheat crop, flowering and grain filling, which are crucial for yield formation. Plant water status indicators, including RWC, MSI, and stomatal conductance (gs) were measured. These parameters served as physiological indicators to the level of stress under the drought treatment. The irrigation system used was surface irrigation following the standard practices for the region. Split-plot design was applied in three replicates in a randomized block arrangement. The irrigation treatments were assigned to the main plots, while the wheat genotypes were allocated in the sub-plots. Each plot contained six rows, each three meters long, with a 20 cm spacing between the rows. Before sowing, a single dose of 35 kg P ha$^{-1}$ phosphorus fertilizer was applied. Three separate applications of nitrogen fertilizer were added at sowing, 30 days following sowing and the tillering stage, with a total amount of 180 kg N/ha. Weeding operation was performed to maintain proper field conditions at 30 days after sowing. Atlantis (Cyflufenamid), a post-emergence herbicide, was applied to control weeds targeting broadleaf and grassy weeds during the tillering stage.

## Measured traits

### Chlorophyll content and photosynthetic efficiency

Fifty-five days after sowing, physiological indicators were assessed on the sixth-node leaf. Concentrations of Chl *a* and *b* were determined by homogenizing 0.5 g of the youngest fully expanded leaf tissue in five mL of cold 85% acetone, followed by centrifugation. Following *Lichtenthaler (1987)*, the optical density was measured using spectrophotometry at 663 and 647 nm after the resulting extract was diluted to the appropriate volume. A portable steady-state parameter (LI-1600, LICOR, Lincoln, NE, USA) was utilized to determine photosynthetic parameters; net photosynthetic rate (NPR), Tr, and gs. To ensure the precision of gs was recorded on a fully expanded flag leaf using three replicates per leaf. The formula A = A$_{max}$ × f(PAR), where A$_{max}$ is the maximum theoretical photosynthetic rate and f(PAR) is a function of photosynthetically active radiation (PAR) recorded using a calibrated quantum sensor at three different times (morning, noon, and afternoon), was

utilized to calculate NPR (*Sicher & Barnaby, 2012*). The same leaves were used to measure Tr, considering both the adaxial and abaxial surfaces directly.

### Water relations and malondialdehyde

Relative water content was determined using the method of *Barrs & Weatherley (1962)*. Fresh weight (FW) was determined from leaves, then immersed in water for 5 h and the turgid weights (TW) were calculated. Then samples were dried in the oven at 80 °C for 24 h and dry weight (DW) was recorded. The RWC was estimated as follows: RWC = ((FW−DE)/(TW−DE))×100.

The youngest leaf tissues (0.2 g) were cut into one cm pieces after cleaning with deionized water and submerged in 10 mL of deionized water, which were utilized to calculate the MSI. After that, these samples were cooked for 30 min at 40 °C in a water bath. Then, a conductivity meter (ME977-C, Max Electronics, Mineola, NY, USA) was used to reorder the electrical conductivity ($EC_1$). The electrical conductivity ($EC_2$) of the samples was then determined after they had been cooked for ten minutes in a water bath at 100 °C (*Premachandra, Saneoka & Ogata, 1990*). To calculate the MSI, the following formula was applied: MSI (%) = [1 −($EC_1$/$EC_2$)] ×100.

Lipid peroxidation was determined by assessing the level of malondialdehyde (MDA) using the protocol of *Hodges et al. (1999)*. Fresh leaf samples (0.1 g) were finely chopped and homogenized in 1.5 mL of 5% trichloroacetic acid (TCA) on ice. The homogenate was centrifuged at 12,000 rpm for 15 min at 25 °C, and the supernatant will be collected for MDA quantification. Equal volumes of the supernatant and 20% TCA containing 0.5% thiobarbituric acid (TBA) were mixed and incubated at 95 °C for 30 min, followed by rapid cooling in an ice bath. The reaction mixture was centrifuged at 1,000 rpm for 5 min, and absorbance was measured at 532 and 600 nm using a spectrophotometer. A TCA-TBA solution will serve as a blank. MDA content (nmol $gFW^{-1}$) will be calculated using the formula:

MDA content = [(A532-A600)×extraction volume]/[155×sample quantity].

### Proline and enzymatic antioxidants activities

Proline content was measured utilizing the procedure outlined in *Bates, Waldren & Teare (1973)*. The procedure involved extracting 0.5 g of plant tissue in 5% sulfosalicylic acid and centrifuging it for 7 min at 10,000×g. The resultant supernatant was boiled for 30 min at 94 °C after being diluted with water and combined with 2% ninhydrin reagent. Toluene was added to the mixture after it had cooled, and the upper organic phase was examined using spectrophotometry at 520 nm. To determine antioxidant enzyme activity, 200 mg of leaf samples were quickly frozen in liquid nitrogen and then crushed in 2.0 mL of an extraction buffer that contained 10 mM ascorbic acid, 100 mM potassium phosphate (pH 7.8) and 0.1 mM ethylenediaminetetraacetic acid (EDTA). The final homogenate was centrifuged for 15 min at 4 °C at 13,000× g. Protein content and enzyme activity were then measured using the supernatant left over after centrifugation. According to *Aebi (1984)*, CAT activity was measured in the supernatant at 240 nm based on $H_2O_2$ consumption. Monitoring the drop in absorbance at 290 nm allowed for measuring APX activity following *Ma &*

**Table 1** Drought tolerance indices, formula equations, and references.

| Drought tolerance indices | Formula equations | References |
|---|---|---|
| Mean productivity (MP) | $(Y_S + Y_p) / 2$ | *Hossain et al. (1990)* |
| Geometric mean productivity (GMP) | $(Y_p)^{(1/2)} \times Y_S$ | *Fernandez (1992)* |
| Stress tolerance index (STI) | $(Y_s \times Y_p)/(Y_p)^2$ | *Fernandez (1992)* |
| Yield index (YI) | $Y_s/Y_p$ | *Gavuzzi et al. (1997)* |

Notes.

Y is grain yield under water stress, $Y_p$ is the grain yield under non-water-stressed conditions, $Y_s$ is the average grain yield of all genotypes under water stress.

*Cheng (2004)*. Using the approach of *Giannopolitis & Ries (1977)*, the SOD activity was determined at 560 nm.

### Agronomic traits

Plant height (cm) was recorded as the distance from the soil surface to the tip of the spike, excluding the awns. Number of grains per spike and spike length (cm) were recorded from ten randomly selected main spikes from each plot. Thousand kernel weight (g) was evaluated as the weight of 1,000 grains. All plants from each plot were harvested, threshed, and the grains were weighed. The grain yield was then calculated by converting the weight to tons per hectare (ton ha$^{-1}$).

## Statistical analysis

The statistical analyses were performed using R software 4.1.1. To assess the normality of the data, the Shapiro–Wilk test was used (*Shapiro & Wilk, 1965*). The Bartlett's test was applied to check for homogeneity of variances across groups, as this is a prerequisite for performing ANOVA (*Bartlett, 1937*). If the data violated assumptions of normality or homogeneity, non-parametric tests, such as the Kruskal–Wallis test, were used instead. To assess differences between the irrigation regime, genotype, and their interaction, the least significant difference (LSD) test ($p < 0.05$ and $<0.01$) was used. The FactoExtra package was employed to perform PCA, and the ComplexHeatmap package was utilized to create a heatmap. Using the procedures outlined by *Burton & De Vane (1953)*, the genotypic and phenotypic variance components and their coefficients of variation were calculated. Drought tolerance indices (Table 1) were computed to classify the genotypes according to their drought tolerance according to *Fernandez (1992)*, *Gavuzzi et al. (1997)*, *Hossain et al. (1990)*. Hierarchical cluster analysis was conducted to group the evaluated genotypes based on their level of drought tolerance using drought tolerance indices as the criteria.

## RESULTS

## Analysis of variance

Combined variance analysis revealed that genotypes and irrigation exhibited highly significant effects on all physiological and agronomic characteristics, highlighting the substantial role of genetic makeup and environmental factors on these traits (Table 2). Significant interactions between genotypes and irrigation were detected across all evaluated characters, underscoring the necessity of studying genotype-by-environment interactions

**Table 2  Analysis of variance (mean squares) for evaluated characters of the assessed genotypes under well watered and drought stress conditions over the two seasons of 2021/2022 and 2022/2023.**

| Source of Variance | df | Chl*a*§ | Chl*b* | NPR | Tr | gs | RWC | MSI | MDA |
|---|---|---|---|---|---|---|---|---|---|
| Growing season (Gs) | 1 | 1.15* | 0.74 | 24.18 | 1.54 | 0.17* | 780.2 | 411.3 | 692.6 |
| Replication/(Gs) | 4 | 0.130 | 0.157 | 9.69 | 0.79 | 0.02 | 67.18 | 96.94 | 95.61 |
| Irrigation (Ir) | 1 | 41.38** | 9.89** | 947.0** | 40.80** | 1.19** | 15,676** | 4,492** | 13,021** |
| Ir×GS | 1 | 0.57 | 0.58 | 12.39 | 0.85* | 0.001 | 72.32 | 13.20 | 22.84 |
| Error a | 4 | 0.29 | 0.11 | 3.43 | 0.09 | 0.001 | 59.86 | 21.10 | 4.58 |
| Genotypes (Gen) | 13 | 0.07** | 0.93** | 53.03** | 1.72** | 0.10** | 106.9** | 204.7** | 191.1** |
| Gen×Gs | 13 | 0.03* | 0.28** | 12.47** | 2.04** | 0.02** | 14.02 | 69.17** | 15.07** |
| Gen×Ir | 13 | 0.08** | 0.26** | 20.59** | 0.91** | 0.06** | 77.79** | 118.8** | 138.8** |
| Gen×Gs×Ir | 13 | 0.04** | 0.30** | 6.22** | 1.14** | 0.01** | 15.73 | 81.55** | 24.05** |
| Error b | 104 | 0.01 | 0.01 | 0.50 | 0.10 | 0.001 | 12.35 | 9.74 | 4.54 |

| Source of Variance | df | Proc | CAT | APX | SOD | PH | NGS | TGW | GY |
|---|---|---|---|---|---|---|---|---|---|
| Growing season (Gs) | 1 | 0.97 | 36.79** | 38.96* | 1,318** | 407.2 | 40.43 | 746.9** | 4.74* |
| Replication/(Gs) | 4 | 0.34 | 1.52 | 2.135 | 3.62 | 185.2 | 5.323 | 8.92 | 0.517 |
| Irrigation (Ir) | 1 | 51.52** | 1,301** | 1,519** | 28,529** | 15,016** | 4,939** | 5,332** | 123.2** |
| Ir×GS | 1 | 4.16* | 47.58** | 44.95** | 312.2* | 51.19 | 53.72* | 281.9 | 34.63* |
| Error a | 4 | 0.33 | 0.55 | 0.51 | 16.08 | 306.6 | 3.10 | 101.9 | 2.05 |
| Genotypes (Gen) | 13 | 0.09** | 20.96** | 37.80** | 84.74** | 1,077** | 419.8** | 103.1** | 6.54** |
| Gen×Gs | 13 | 0.04** | 0.72 | 10.71** | 44.71** | 54.78** | 41.94** | 66.94** | 0.45 |
| Gen×Ir | 13 | 0.10** | 11.85** | 47.91** | 55.97** | 28.73* | 101.0** | 36.05** | 1.91** |
| Gen×Gs×Ir | 13 | 0.04** | 1.03 | 7.68** | 33.82** | 40.58** | 52.74** | 13.24 | 0.35 |
| Error b | 104 | 0.001 | 0.70 | 1.54 | 5.83 | 15.68 | 11.38 | 14.63 | 0.80 |

**Notes.**

§ Chl*a*, Chlorophyll *a* (mg/g FW); Chl*b*, Chlorophyll *b* (mg/g FW); NPR, Net photosynthetic rate ($\mu$mol $CO_2$/m$^2$/s; Tr, Transpiration rate ($\mu$mol $CO_2$/m$^2$/s); gs, Stomatal conductance ($\mu$mol $CO_2$/m$^2$/s); RWC, Relative water content (%); MSI, Membrane stability index (%); MDA, Malondialdehyde ($\mu$mol/g FW); Proc, Proline content ($\mu$mol/g DW); SOD, Superoxide dismutase (unit mg/protein); CAT, Catalase (unit mg/protein); APX, Ascorbate peroxidase (unit mg/protein); NGS, Number of grains/spike; PH, Plant height (cm); TGW, 1000-grain weight (g); and GY, Grain yield (tons/ha).

* and ** Indicate *p*-value <0.05 and 0.01 in the same order.

when developing strategies for crop improvement. The three-way interaction between genotypes, irrigation, and growing seasons was insignificant for grain yield and most assessed characters.

Sixteen physiological and agronomic characters were studied under well-watered and drought-stressed conditions. Significant variations were observed across all traits between irrigation treatments. Under drought conditions, a substantial reduction was observed in Chl*a*, Chl*b*, Tr, NPR, gs, RWC, MSI, plant height (PH), number of grains per spike (NGPS), and GY (Fig. 1). In contrast, traits such as MDA, Proc, CAT, APX, and SOD showed significant increases under drought conditions, suggesting their role in drought tolerance mechanisms. The genotypes exhibited substantial differences in all characters under well-watered and drought conditions, emphasizing genetic variability and its potential for breeding programs.

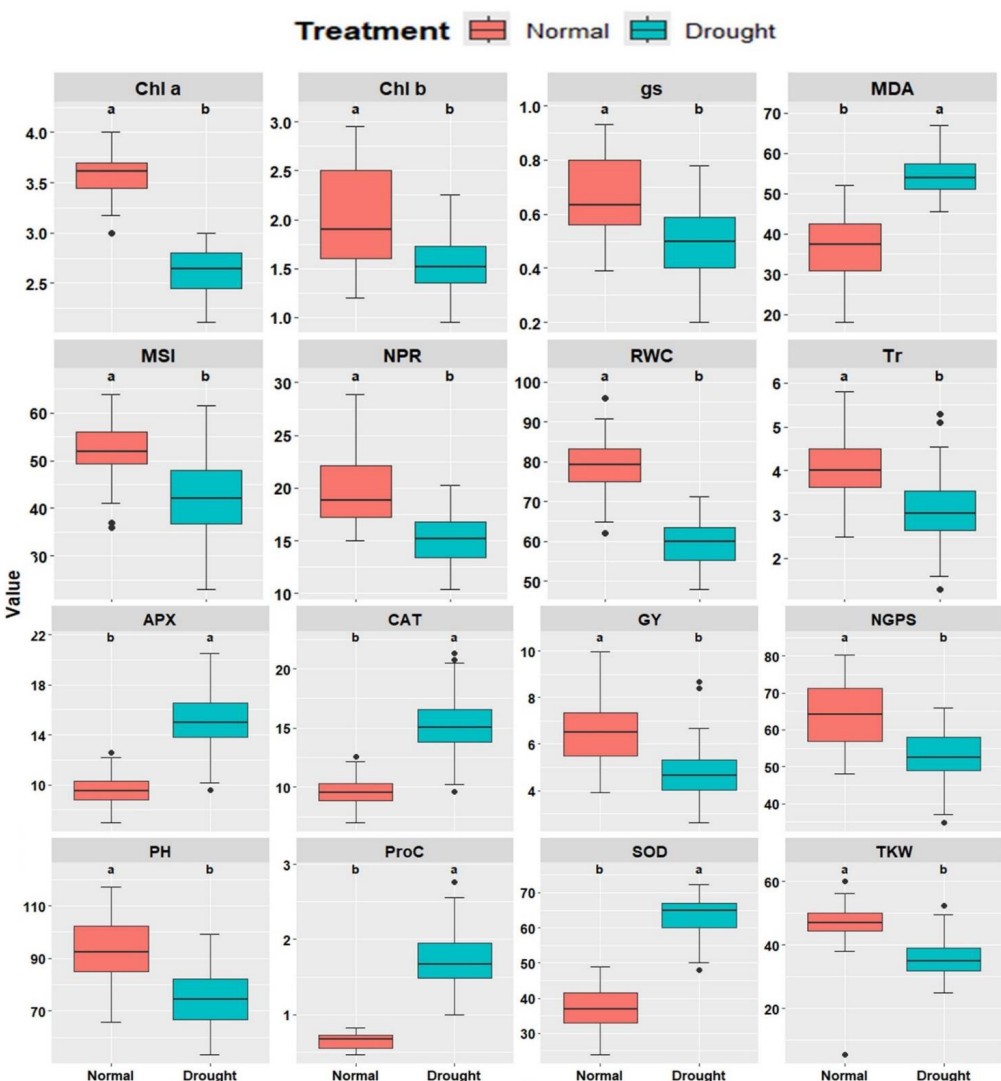

**Figure 1 Comparative boxplots of 16 physiological and agronomic measured traits under normal and drought conditions.** Chlorophyll *a* (mg/g FW), Chl*b*, Chlorophyll *b* (mg/g FW); NPR, Net photosynthetic rate ($\mu$mol $CO_2$/m 2/s; Tr, Transpiration rate ($\mu$mol $CO_2$/m 2/s); gs, Stomatal conductance ($\mu$mol $CO_2$/m 2/s); RWC, Relative water content (%); MSI, Membrane stability index (%); MDA, Malondialdehyde ($\mu$mol/g FW); Proc, Proline content ($\mu$mol/g DW); SOD, Superoxide dismutase (unit mg/ protein); CAT, Catalase (unit mg/ protein); APX, Ascorbate peroxidase (unit mg/ protein); PH, Plant height (cm); NGS, Number of grains /spike; TGW, 1000-grain weight (g); and GY, Grain yield (tons/ha).

## Physio-biochemical attributes

The physio-biochemical attributes of the fourteen evaluated wheat genotypes under well-watered and drought conditions are presented in Figs. S2–S4. Chl *a* decreased under drought from 3.48–3.77 mg/g FW to 2.36–2.81 mg/g FW, with G10, G11, G12, and G8 maintaining the highest levels. Similarly, Chl *b* declined from 1.49–2.65 mg/g FW to 1.32–1.88 mg/g FW, with G12, G13, G11, G8, and G3 exhibiting superior content. NPR dropped from 16.48–25.75 g/m$^2$ to 12.02–17.44 g/m$^2$, with G1, G8, G6, and G9 maintaining

the highest NPR. Tr declined from 3.57–4.56 $\mu$mol $H_2O$/m$^2$/s to 2.17–4.04 $\mu$mol $CO_2$/m$^2$/s, where G2, G14, G5, and G7 recorded the highest values. gs ranged from 0.48–0.90 $\mu$mol $CO_2$/m$^2$/s under well-watered conditions but decreased to 0.34–0.66 cm$^2$/s under drought, with G3, G11, G6, and G8 maintaining higher values. RWC dropped from 69.94%–86.35% to 54.76%–64.54%, where G8, G3, G12, and G13 exhibited superior retention. MSI declined from 46.50%–60.94% to 32.83%–51.96%, with G3, G4, G12, and G8 demonstrating better stability. Conversely, MDA increased from 23.19–46.89 $\mu$mol/g FW under well-watered conditions to 49.93–60.09 $\mu$mol/g FW under drought, with G1, G8, and G9 showing the lowest oxidative stress levels. Proline content increased from 0.58–0.73 $\mu$mol/g FW to 1.54–2.15 $\mu$mol/g FW under drought, with G8, G3, G10, G13, and G12 exhibiting the highest content. Antioxidant enzyme activities were also elevated under drought. CAT increased from 7.63–12.03 units/mg protein to 10.79–18.69 units/mg protein, with G5, G6, G8, and G12 demonstrating the highest activity. APX rose from 4.88–9.62 units/mg protein to 7.73–18.72 units/mg protein, where G8, G12, G6, and G2 excelled. SOD increased from 33.29–40.20 units/mg protein to 53.90–68.59 units/mg protein, with G8, G10, G3, and G2 exhibiting superior activity. These findings highlight the physiological and biochemical adaptations of specific genotypes to drought stress, supporting their potential in breeding programs for improved stress resilience.

## Agronomic traits

Plant height ranged from 79.21 to 108.17 cm in well-watered conditions, with G3, G8, G4 and G6 produced the tallest plants and decreased to 62.29 to 90.02 cm under drought, where G3, G4, G8 and G1 performed best, while G14, G13, and G11 had the shortest plants (Fig. S4A). Number of grains/spike fluctuated from 53.86 to 74.72 in well-watered conditions, with G4, G3, G11, and G8 achieving the highest value, and dropped to 40.74 to 61.02 under drought, with G3, G8, G6, and G10 showing the best performance (Fig. S4B). Thousand kernel weight varied from 40.92 to 54.55 g in well-watered conditions, with G4, G8, G6, G5, and G12 producing the heaviest kernels, and decreased to 31.89 to 41.67 g under drought, where G8, G3, and G1 exhibited superiority (Fig. S4C). Grain yield (GY) ranged from 5.14 to 8.22 tons/ha in well-watered conditions, with G3, G4, G8, and G1 achieving the highest yield and reduced to 3.51 to 6.32 tons/ha under drought, where G8, G12, G3, and G6 outperformed other genotypes (Fig. S4D).

## Genotypic classification

The heatmap analysis revealed three distinct groups among the 14 bread wheat genotypes (G1–G14) based on physio-biochemical and agronomic traits under drought conditions (Fig. 2). Cluster 1 (G5, G7, G2, and G14) demonstrated high levels of Tr and malondialdehyde content. Cluster 2 (G3, G8, G12) is characterized by high chlorophyll content and photosynthetic rate indicating strong photosynthetic efficiency, also high proline levels and antioxidant enzyme activities (APX, CAT, SOD), making these genotypes highly resilient to drought conditions. Besides, this cluster displayed high grain yield and its contributing traits. Cluster 3 (G1, G4, G6, G9, G13, G10, G11) showed moderate levels across most traits. Notably, G3, G8, and G12 could be considered as the most resilient
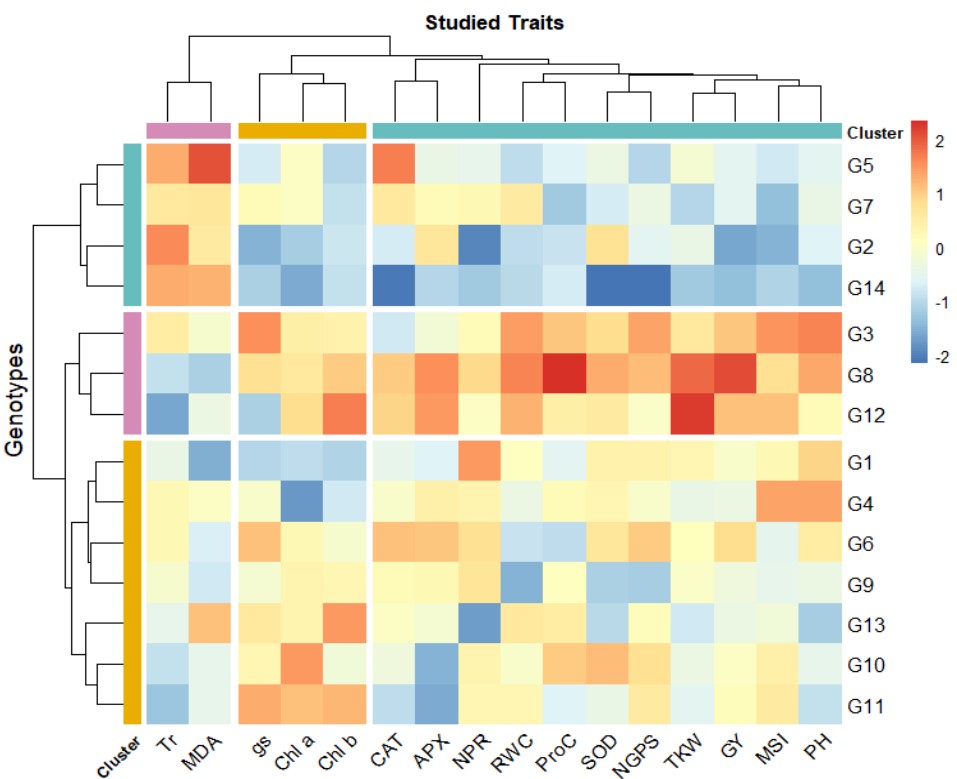

**Figure 2 Heatmap for wheat genotypes based on studied physiological and agronomic traits under drought stress.** Blue color indicates lower rank of studied character while red color indicates high rank. Chlorophyll *a* (mg/g FW), Chl*b*, Chlorophyll *b* (mg/g FW); NPR, Net photosynthetic rate ($\mu$mol $CO_2$/m 2/s; Tr, Transpiration rate ($\mu$mol $CO_2$/m 2/s); gs, Stomatal conductance ($\mu$mol $CO_2$/m 2/s); RWC, Relative water content (%); MSI, Membrane stability index (%); MDA, Malondialdehyde ($\mu$mol/g FW); Proc, Proline content ($\mu$mol/g DW); SOD, Superoxide dismutase (unit mg/ protein); CAT, Catalase (unit mg/ protein); APX, Ascorbate peroxidase (unit mg/ protein); PH, Plant height (cm); NGS, Number of grains /spike; TGW, 1000-grain weight (g); and GY, Grain yield (tons/ha).

genotypes, integrating strong physiological, biochemical, and agronomic responses, while G2, G5, G7, and G14 may require optimal conditions to maximize yield. The clustering also emphasized the association of chlorophyll content, antioxidant activity, and RWC with drought tolerance. Additionally, yield-related traits (GY, NGPS, thousand kernel weight (TKW)) were closely associated with key physiological parameters such as RWC, MSI, NPR, SOD, and proline content, reinforcing their role in genotype differentiation under water deficit stress.

## Drought tolerance indices

The assessed genotypes were further classified using the computed drought tolerance indices, stress tolerance index, geometric mean productivity, mean productivity, harmonic mean, and yield index (Table S3). The hierarchical clustering analysis categorized the genotypes into four groups (Fig. 3). Group A contained three genotypes (G8, G3, and

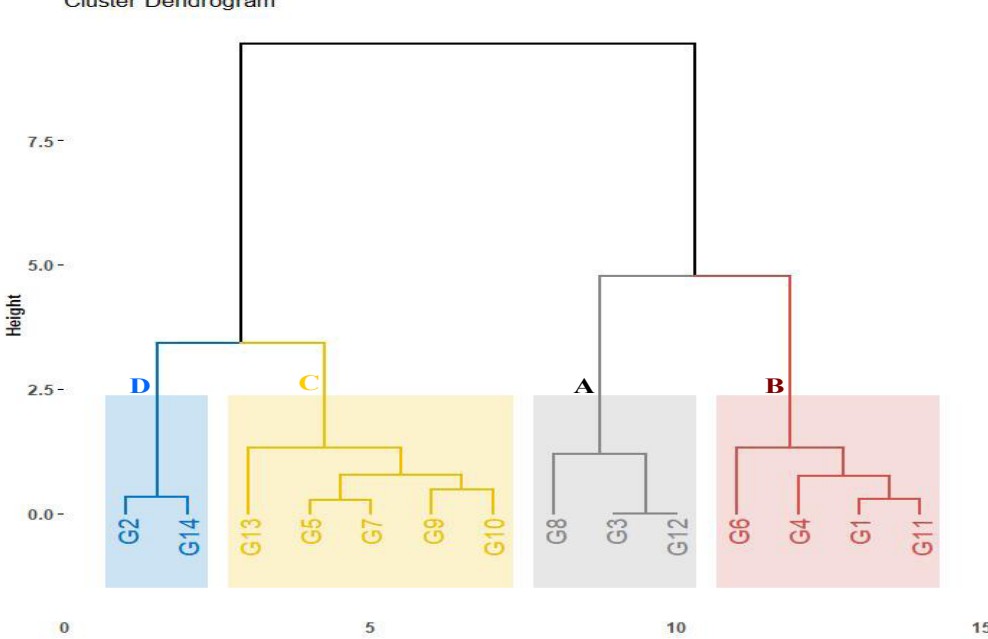

**Figure 3** Dendrogram depicting distances among fourteen wheat genotypes according to tolerance indices.

G12) that exhibited the highest tolerance indices, classifying them as drought-tolerant-genotypes. These genotypes exhibited strong adaptability to drought conditions and represent valuable candidates for wheat breeding to enhance drought tolerance. Group B comprised four genotypes (G6, G4, G1, and G11) demonstrating moderate tolerance index values. These genotypes displayed intermediate performance under drought stress. In contrast, five genotypes (G13, G10, G9, G5, and G7) in Group C, followed by two genotypes (G2 and G14) in Group D that displayed the lowest tolerance index, were identified as drought-sensitive genotypes.

## Ranking and AMMI analyses

The ranking biplot assessed the stability and performance of 14 wheat genotypes (G1–G14) across four distinct environments (E1–E4) under well-watered and drought conditions over two seasons (Fig. 4A). PC1 accounted for 74.07% of the total variation, while PC2 explained 19.58% of the genotype-by-environment interaction variance. The average environment coordinate (AEC) was used to evaluate genotype performance and stability, where genotypes positioned closer to the AEC and aligned along its axis exhibited higher performance and stability. Among these, G6, G3, G8, and G12 demonstrated superior stability and broad adaptability, making them ideal for cultivation under both well-watered and drought conditions. In contrast, genotypes G2, G14, and G13, which were farther from the AEC, performed well in specific environments, suggesting their suitability for favorable growing conditions rather than general adaptability. The environments also displayed variability in their alignment with the AEC, reflecting distinct characteristics influencing genotype

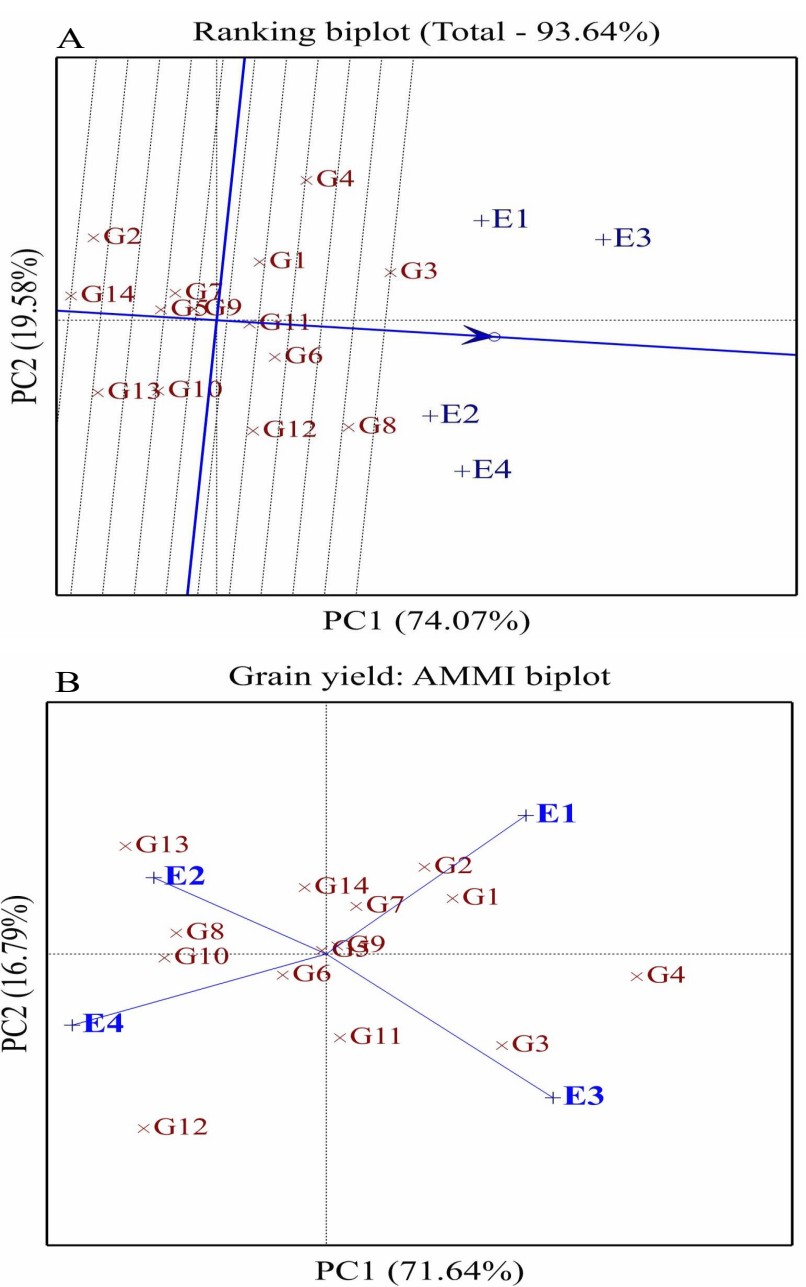

**Figure 4** **Comparison of fourteen wheat genotype performance (G1-G14) and stability across environments using ranking (A) and AMMI biplots (B) for grain yield.** E1: Normal conditions during the first season, E2: Drought conditions during the first season, E3: Normal conditions during the second season, and E4: Drought conditions during the second season.

performance. Well-watered environments E1 and E3 were positioned on the positive side of PC1, aligning with high-yielding genotypes. The AMMI analysis further highlighted unique genotype-environment interactions. E1, representing favorable well-watered conditions, was located in the positive PC1 and PC2 sector, where G1 and G2 performed well. E2, representing drought conditions in the first season, was situated in the positive PC2 and negative PC1 sector, favoring G13 and G8 (Fig. 4B). E3, corresponding to well-watered conditions in the second season, was positioned in the positive PC1 and negative PC2 sector, aligning with G3 strong performance. E4, under drought conditions in the second season, was located in the negative PC1 and PC2 sector, where G12 performed well. Some genotypes, such as G5, G6, G9, and G7, were positioned near the biplot origin, indicating minimal interaction effects and broad adaptability, making them suitable for cultivation across diverse environments.

## Association analyses

Principal component analysis was applied to explore the relationships among studied physiological and agronomic characters. The biplot (Fig. 5A) segregated the well-watered and drought conditions indicating diverse associations. The first two PCs (PC1 and PC2) explained 79.94% of the total variance, with PC1 accounting for 69.42% and PC2 for 10.52% (Fig. 5B). Traits such as APX, MDA, SOD, Proc, and CAT were associated with drought stress were presented in distinct blue cluster. In contrast, traits related to well-watered conditions were presented in the separated yellow cluster, indicating to their differing responses to environmental conditions. Regarding trait contributions, Chl *a* had the highest contribution to PC1 (8.23%), followed by RWC (8.0%) and Proc (7.56%). For PC2, Tr was the dominant contributor (22.5%), followed by GY (11.32%) (Figs. 5B, 5C).

Spearman correlation matrix for physiological and agronomic parameters under well-watered and drought conditions is presented in Figs. 6A and 6B. The correlation analysis revealed different patterns under well-watered and drought conditions. Under well-watered conditions (Fig. 6A), positive associations were detected between grain yield and several key characters, including Chl *a*, MSI, NPR, PH, NGPS, and TKW. These associations emphasized the significance of these characters in determining yield under well-watered conditions. Besides, Chl *a* exhibited a strong positive association with NPR, PH, and NGPS, indicating its role in influencing agronomic performance. Other positive associations included NPR with SOD and RWC with APX, suggesting that improved photosynthesis and water retention are associated with enhanced antioxidant activity. Conversely, Tr displayed a strong negative association with NPR and MDA, indicating that oxidative stress negatively impacts photosynthetic efficiency. Under drought conditions (Fig. 6B), GY exhibited significant positive associations with Chl *a*, Chl *b*, NPR, MSI, RWC, ProC, SOD, NGPS, and TKW. Also, TKW was positively associated with RWC, ProC, and APX. In addition, NGPS showed strong positive correlations with PH, RWC, and SOD. Other pairwise associations were identified, such as Chl *a* with Chl *b*, CAT with APX, ProC with SOD, and RWC with CAT. Negative correlations were also observed between GY and MDA.

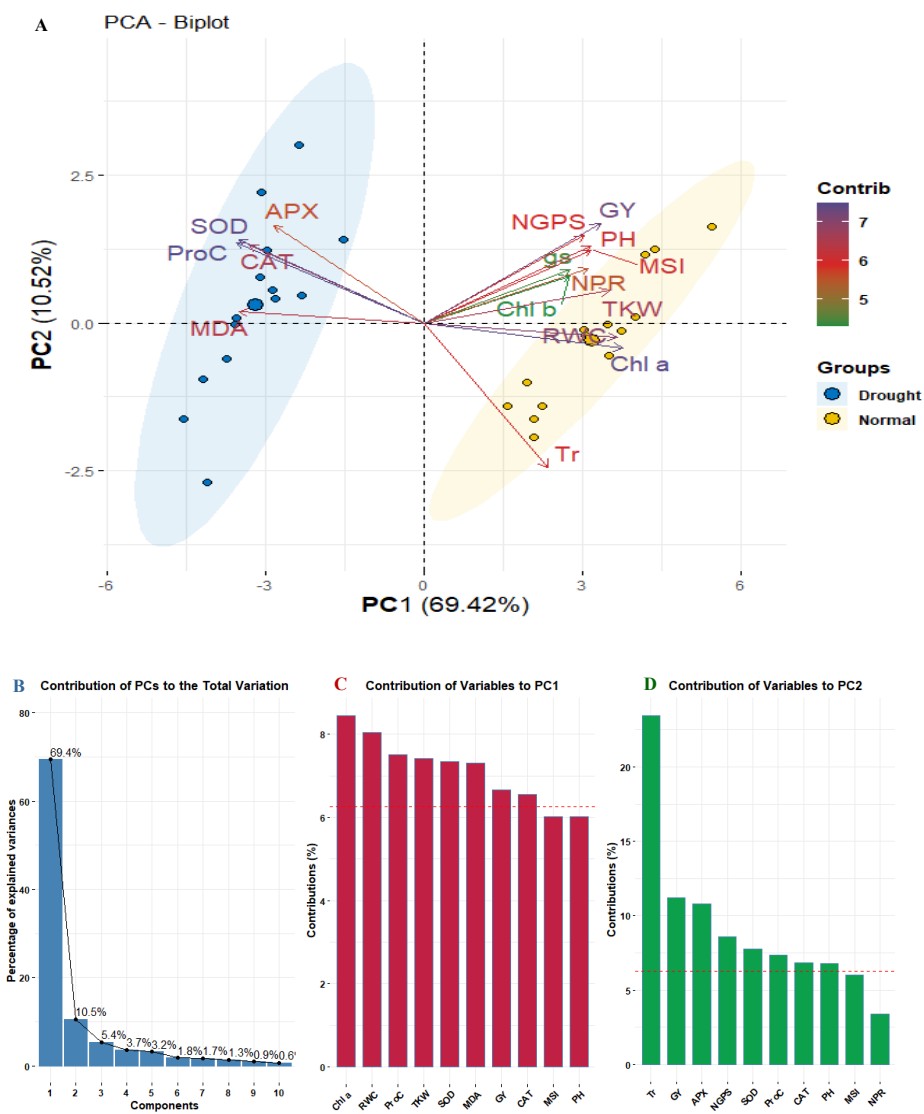

**Figure 5** PCA biplot for studied physiological and agronomic in fourteen wheat genotypes under normal and water deficit conditions (A), bar chart with contribution percentage of principal components to overall variance (B), and bar charts of trait contribution with dashed line of significant contribution (C and D).

## Path analysis

The path analyses revealed key physiological and agronomic traits influencing grain yield under drought conditions. The studied parameters RWC, MSI, Chl *a* and Chl *b*, CAT, APX, and SOD exhibited strong positive direct effects on grain yield, indicating their critical role in drought adaptation (Table S4). In contrast, MDA had a negative direct impact, suggesting that oxidative stress impairs yield performance. In addition to direct contributions, several traits influenced grain yield indirectly. RWC enhanced yield by promoting photosynthetic efficiency and antioxidant enzyme activity, while antioxidant enzymes (CAT, APX, and

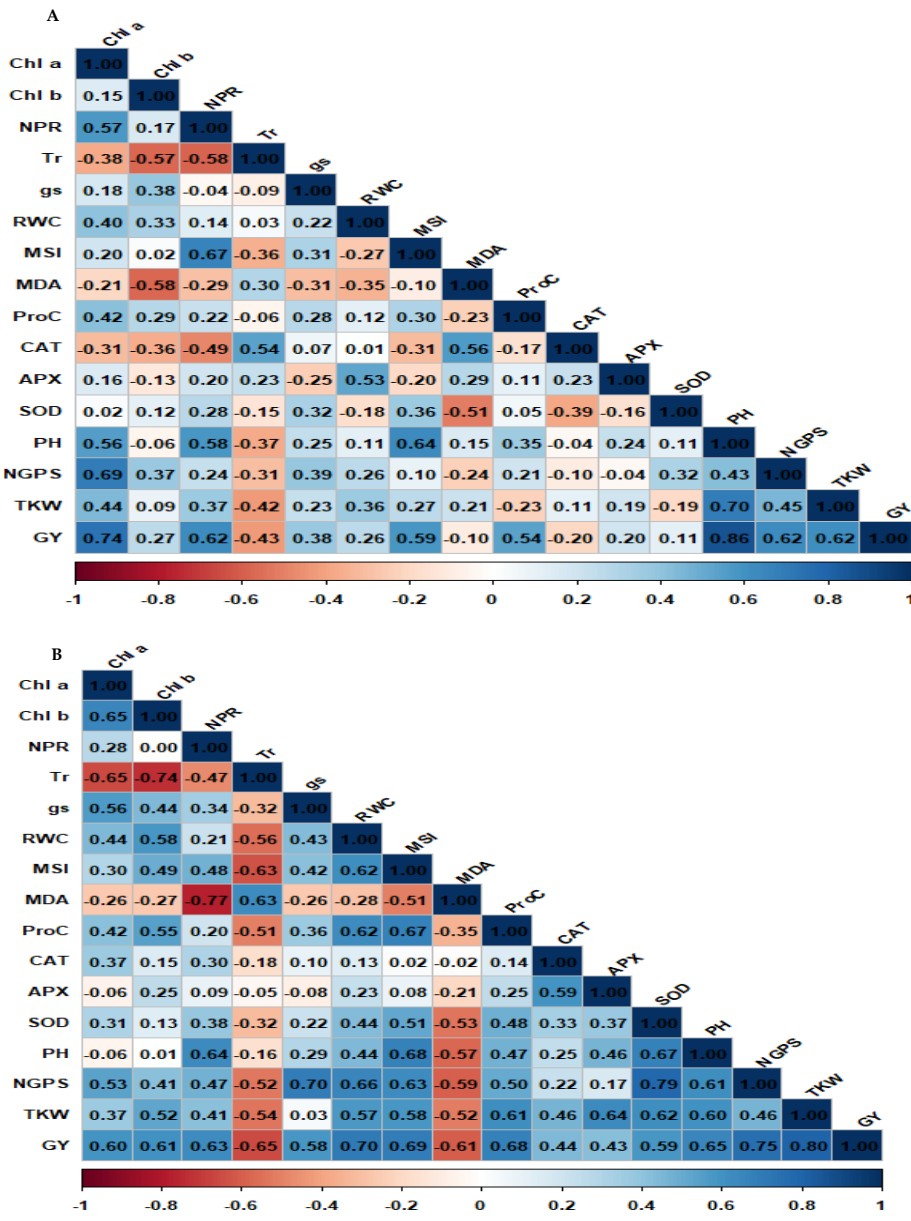

**Figure 6  Matrix of Spearman correlation for physiological and agronomic characters under well-watered (A) and water deficit conditions (B).**

SOD) contributed indirectly by improving RWC and MSI, thereby mitigating oxidative stress. Among agronomic traits, TKW and PH positively influenced grain yield, primarily by supporting physiological traits such as MSI and chlorophyll content. These findings indicated the biological significance of maintaining water status, antioxidant defense, and photosynthetic efficiency in ensuring grain productivity under water deficit conditions.

## Heritability, variance component, and genetic advance

The phenotypic variance exceeded the genotypic variance in all studied traits under well-watered conditions (Table S5). Among the characteristics, PH and NGPS had the greatest values of GCV and PCV, while the lowest values were observed for Chl*a*, Proc, and gs. High PCV values were recorded for NGPS, PH, MSI, and TKW. In contrast, GY, Tr, and Proc exhibited moderate PCV values. Likewise, GCV was highest for APX, NPR, and MSI, while PH, TKW, Tr, and gs displayed moderate GCV values. Broad-sense heritability estimates varied from moderate to high across the studied characters. GY and Proc exhibited moderate genetic advance and heritability. In contrast, NPR, SOD, APX, and PH displayed high heritability with genetic advance. Similarly, phenotypic variance exceeded genotypic variance for all characters under drought stress. PCV and GCV values were uppermost for APX, MDA, and NPR, while the lowest values were observed for Chl *a*, gs, and GY. NGPS, PH, MSI, and TKW exhibited high PCV values, whereas moderate PCV values were recorded for Tr, gs, and Proc. Likewise, GCV was highest for APX, NPR, and MDA, while moderate GCV values were observed for PH, TKW, Tr, and GY. Broad-sense heritability varied from moderate to high for the studied characters. APX, SOD, MDA, and NGPS exhibited high heritability and genetic advance under drought stress. In contrast, GY, gs, and Proc showed moderate heritability and genetic advance.

## DISCUSSION

Climate change increases drought stress in arid Mediterranean regions and severely diminishes wheat yields (*Melki et al., 2024*). Therefore, developing drought-tolerant genotypes is essential for sustaining wheat production. Screening genotypes for tolerance to water scarcity using physiological and agronomic traits under natural field conditions in targeted environments is an effective strategy for enhancing wheat breeding in arid environments. This study focused on physiological and agronomic performance of diverse bread wheat genotypes under drought-stressed and non-stressed conditions in an arid environment. The ANOVA results demonstrated that both genotype and irrigation had a significant impact on all studied traits. These results showed that the assessed materials possessed high degree of genetic variability, which could be employed in improving wheat productivity. These results are in consonance with *Morsy et al. (2022)*, *Saidi et al. (2024)*, *Tefera et al. (2021)*, who reported genetic variability among genotypes for physiological and agronomic characters under stressed and well-watered conditions. The observed significant genotype × irrigation interaction across studied traits highlighted the varied responses of genotypes to irrigation treatments. This result indicated the importance of studying these interactions in wheat breeding to develop resilient genotypes to diverse environmental conditions. Similarly, *Kutlu et al. (2021)*, *Ru et al. (2024)*, *Upadhyay et al. (2023)* depicted the vital effects of genotype-environment interactions on the expression of physiological and agronomic characters.

The assessed genotypes displayed different responses to drought stress. The genotypes were classified into different groups ranging from drought sensitive to tolerant. Genotypes G3, G8, and G12 demonstrated superior performance under drought conditions,

characterized by higher Chl *a* and *b*, RWC, and enhanced antioxidant enzyme activities (CAT, APX, and SOD). These parameters displayed the ability of these genotypes to sustain photosynthetic efficiency and cellular stability, contributing to higher grain yields under drought stress. Moreover, elevated proline and antioxidant levels emphasized their superior drought tolerance, because these characters are crucial in osmotic adjustment and scavenging ROS. The strong performance under drought stress identifies these genotypes as promising candidates for wheat breeding to enhance its productivity under limited water conditions (*Ahmad et al., 2022*). The drought tolerance indices and ranking analyses revealed distinct performance patterns among the 14 wheat genotypes across varying environments. Notably, G3, G6, G8, and G12 were considered as the most stable and high-performing genotypes, consistently aligning with the AEC in the ranking biplot. Their proximity to the AEC axis signifies broad adaptability and resilience across both well-watered and drought-stressed conditions, making them ideal candidates for breeding and large-scale cultivation in variable climates. In contrast, genotypes G2, G14, and G13, which deviated significantly from the AEC, demonstrated environment-specific performance, favoring favorable growing conditions. The AMMI biplot analysis further highlighted key genotype-environment interactions (*El-Abssi et al., 2024*; *Mansour et al., 2018a*; *Megahed et al., 2022*). G12 showed strong association with E4 (drought conditions in the second season), indicating its ability to maintain performance under water deficit stress. Similarly, G8 and G13 were linked to E2 (drought in the first season), suggesting their drought resilience during early-season stress. In contrast, G1 and G2, associated with E1 (well-watered conditions in the first season), performed well under optimal conditions but lacked consistency under drought stress. G5, G6, G7, and G9, located near the biplot origin, exhibited minimal interaction effects, indicating broad adaptability across environments. The PCA reinforced these findings and effectively distinguished the assessed genotypes based on their adaptability. The separation of drought-prone environments (E2 and E4) on the negative side of PC1 and well-watered environments (E1 and E3) on the positive side highlights the differential responses of genotypes. These results emphasize the importance of selecting genotypes with both high yield potential and stability across diverse conditions. Therefore, these multivariate analyses are important in differentiating wheat genotypes based on physiological and agronomic traits under stress conditions (*ElShamey et al., 2022*; *Gracia et al., 2012*; *Habibullah et al., 2021*; *Mansour et al., 2018b*; *Ponce-Molina et al., 2012*).

The genotypes used in this study included local and exotic genotypes. Local genotypes have adapted to agroclimatic conditions over time, often exhibiting traits that enhance performance under environmental stresses (*Gharib et al., 2021*; *Mansour et al., 2021*). In contrast, exotic genotypes, while sometimes offering superior yield potential, may lack the same level of adaptation to local environmental challenges (*Morsi et al., 2023*; *Zannat et al., 2023*). The results revealed that some exotic genotypes, particularly G3 and G8, demonstrated high levels of drought resilience, suggesting their potential for introgression into local breeding programs. However, local genotypes such as G6 and G12 also showed strong adaptability, indicating that breeding strategies should focus on incorporating the drought-tolerance traits of both sources to develop superior cultivars. The genetic

diversity between local and exotic genotypes provides an opportunity for hybridization strategies that combine the resilience of local varieties with the high-yield potential of exotic lines. This approach can lead to the development of improved wheat varieties that are both productive and well-adapted to Egypt's challenging growing conditions, ultimately enhancing the efficiency of wheat breeding programs in the region.

The association analyses provided valuable insights into the relationships between yield traits and physiological parameters under different water availability conditions. The PCA results indicated a clear segregation of traits based on well-watered and drought conditions, presenting the distinct physiological responses to environmental stress. The clustering of drought-associated traits (APX, MDA, SOD, ProC, and CAT) in a separate group suggests that oxidative stress response mechanisms play a critical role in adaptation to water deficit. Conversely, traits related to optimal water conditions formed a separate cluster, reinforcing the idea that different physiological pathways are activated depending on water availability. The correlation analysis further validated these findings by demonstrating significant shifts in trait associations between well-watered and drought conditions. Under drought conditions, grain yield exhibited strong positive associations with Chl *a*, Chl *b*, NPR, MSI, RWC, ProC, SOD, NGPS, and TKW, indicating that plants with higher antioxidant enzyme activities and better water retention performed better under water deficit. Additionally, TKW correlated positively with RWC, ProC, and APX, further supporting the role of water retention and oxidative stress mitigation in sustaining grain development. Interestingly, negative correlations between GY and MDA suggest that reducing oxidative damage is crucial for maintaining productivity under stress. Path analysis reinforced these observations by identifying both direct and indirect effects of physiological and agronomic traits on grain yield under drought conditions. Water relations (RWC, MSI), chlorophyll content (Chl *a* and Chl *b*), and antioxidant enzyme activities (CAT, APX, and SOD) had positive direct effects on grain yield, suggesting that maintaining cellular hydration, photosynthetic efficiency, and oxidative stress mitigation are critical determinants of yield stability under drought. Additionally, indirect effects were observed, with RWC enhancing grain yield through improved photosynthetic efficiency and antioxidant activity, while antioxidant enzymes contributed to yield through their role in maintaining MSI and RWC. The agronomic traits, particularly TKW and PH, indirectly influenced grain yield by enhancing physiological parameters like chlorophyll content and MSI. In this regard, the importance of physiological parameters as markers of grain yield under drought stress was clarified by *Desoky et al. (2023)*, *Devi et al. (2024)* and *Pantha et al. (2024)*.

Assessing genetic variability and heritability of physiological and agronomic parameters under well-watered and water scarcity conditions is pivotal for developing drought-tolerant wheat genotypes. In the present study, higher PCV compared to GCV for most characters under well-watered and drought conditions indicated a significant effect of irrigation treatments on studied traits. These results align with the research of *Pour-Aboughadareh et al. (2020)*, *Mansour et al. (2023)*, *Sewore & Abe (2024)*, which also stated significant variability and potential for selection across physiological and agronomic characters under both conditions. Heritability values were slightly higher under drought conditions than

under well-watered conditions for most characters, suggesting that selection for moisture response is more feasible in stress environments (*Beyene et al., 2015*). When combined with substantial genetic advances, high heritability suggests the presence of additive gene effects, indicating that selection could lead to meaningful genetic gains (*Johnson, Robinson & Comstock, 1955*).

The findings of this study have significant practical implications for wheat breeding programs aimed at improving drought tolerance. The substantial genetic variability observed among the tested wheat genotypes indicated the potential for selecting high-yielding, drought-resilient cultivars. Specifically, genotypes G3 (L-1117), G8 (L-120), and G12 (L-1142) demonstrated superior physiological, biochemical, and agronomic traits, making them ideal candidates for breeding programs focused on drought adaptation. The clustering analysis further categorized genotypes based on their drought adaptability, with Group A genotypes (G3, G8, and G12) considered as the most drought-tolerant, confirming their potential use in breeding programs. Moreover, stability analyses revealed that genotypes (L-125), G3 (L-1117), G8 (L-120), and G12 (L-1142) exhibited consistent performance across multiple environments, indicating their broad adaptability. The combination of physiological, biochemical, and agronomic evaluations provides a robust framework for selecting superior wheat lines suitable for cultivation in drought-prone regions, ultimately contributing to food security and sustainable wheat production. The positive associations of grain yield with key physiological parameters such as Chl *a* and Chl *b*, MSI, RWC, and antioxidant enzyme activities (APX, CAT, and SOD) highlight their importance in maintaining plant productivity under water-deficit conditions. These traits can be targeted in breeding programs to enhance drought resilience while sustaining grain yield. Additionally, the identification of stress-responsive traits, such as increased proline content and antioxidant enzyme activity under drought, reinforces their role as reliable indicators for screening drought-tolerant genotypes.

## CONCLUSIONS

The results displayed adverse effects of drought conditions on wheat productivity, with significant reductions in photosynthetic parameters, water retention ability, and yield traits. However, activating defense mechanisms, such as increased proline and antioxidant activities, displayed an important role in mitigating stress impacts. Genotypes G3, G8, and G12 exhibited superior resilience and consistent yield traits under drought stress. Therefore, these genotypes could be considered promising for improving drought resilience and ensuring sustainable wheat production in drought-prone regions. In contrast, the remaining moderately tolerant and sensitive genotypes require targeted improvement. Key traits, including Chl *a* and *b*, RWC, photosynthetic efficiency (NPR, Tr and gs), and antioxidant enzyme activities (CAT, APX, SOD), were identified as crucial indicators of drought tolerance while reducing MDA levels was essential for improving drought tolerance. These traits exhibited high heritability and genetic advance, providing a strong foundation for genetic improvement. Consequently, integrating these biochemical and physiological parameters with agronomic traits in wheat breeding programs could offer

an efficient approach to improve drought tolerance and address the challenges of climate variability.

### Funding

This work was supported by the Deanship of Scientific Research and Graduate studies at King Khalid University through a large group Research Project under grant number RGP2/295/44. Additionally, this publication has been supported the RUDN University Scientific Projects Grant System, project No 202787-2-000. This work was also supported by Princess Nourah bint Abdulrahman University Researchers Supporting Project number (PNURSP2025R318), Princess Nourah bint Abdulrahman University, Riyadh, Saudi Arabia. The funders had no role in study design, data collection and analysis, decision to publish, or preparation of the manuscript.

### Grant Disclosures

The following grant information was disclosed by the authors:
Deanship of Scientific Research and Graduate studies at King Khalid University through a large group Research Project: RGP2/295/44.
RUDN University Scientific Projects Grant System: 202787-2-000.
Princess Nourah bint Abdulrahman University Researchers Supporting Project: PNURSP2025R318.

### Competing Interests

Diaa Abd El-Moneim is an Academic Editor for PeerJ.

### Author Contributions

- Mohammed O. Alshaharni conceived and designed the experiments, analyzed the data, prepared figures and/or tables, authored or reviewed drafts of the article, and approved the final draft.
- Fatmah A. Safhi performed the experiments, analyzed the data, prepared figures and/or tables, and approved the final draft.
- Nora M. Al Aboud conceived and designed the experiments, analyzed the data, prepared figures and/or tables, and approved the final draft.
- Dmitry E. Kucher performed the experiments, authored or reviewed drafts of the article, and approved the final draft.
- Eman Fayad conceived and designed the experiments, performed the experiments, prepared figures and/or tables, and approved the final draft.
- Mohammed Alqurashi performed the experiments, analyzed the data, prepared figures and/or tables, and approved the final draft.
- Rahmah N. Al-Qthanin conceived and designed the experiments, authored or reviewed drafts of the article, and approved the final draft.
- Ibtesam S.M. Almami performed the experiments, prepared figures and/or tables, authored or reviewed drafts of the article, and approved the final draft.

- Heba I. Ghamry conceived and designed the experiments, analyzed the data, prepared figures and/or tables, and approved the final draft.
- Diaa Abd El-Moneim conceived and designed the experiments, performed the experiments, analyzed the data, authored or reviewed drafts of the article, and approved the final draft.
- Mohamed M. Kamara conceived and designed the experiments, analyzed the data, prepared figures and/or tables, authored or reviewed drafts of the article, and approved the final draft.
- Abdelraouf M. Ali conceived and designed the experiments, performed the experiments, analyzed the data, prepared figures and/or tables, authored or reviewed drafts of the article, and approved the final draft.

## Data Availability

The raw data is available in the Supplementary File.

## Supplemental Information

Supplemental information for this article can be found online at http://dx.doi.org/10.7717/peerj.19341#supplemental-information.

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
