# Peer review of "Genetic variability and trait associations for physiological and agronomic characteristics in bread wheat genotypes under drought stress and well-watered conditions"

_PeerJ, doi:10.7717/peerj.19341_

## Round 0.1 · original submission · Major Revisions

Revise your manuscript to improve it, taking into account reviewer comments.

Reviewer 1 ·

Basic reporting

The article is well-written in English but has minor repetitions that were determined.
In lines 167 and 168, the two successive sentences start with "After that,"
Please check again for repeated words and sentences. It needs to be improved in the M&M section.

Experimental design

The experimental design is appropriate to reach the answer to the research question. On the other hand, how drought indices were calculated (if possible with references) should be explained.
The relationship between seed yield and the investigated parameters should be explained in detail.
In the Abstract, drought-tolerant genotype(s) associated with the drought-related traits should be given.

Validity of the findings

The findings are very useful for breeders working on drought tolerance in wheat.

Reviewer 2 ·

Basic reporting

Title can be shortened.
Numerical values, especially of decisive characters, regarding the measured data can be given in the summary section.
Literatures are up to date.

Experimental design

Research question well defined, relevant & meaningful. It is stated how research fills an identified knowledge gap.

Validity of the findings

Conclusions are well stated, linked to original research question & limited to supporting results.

Annotated reviews are not available for download in order to protect the identity of reviewers who chose to remain anonymous.

·

Basic reporting

This manuscript investigates the genetic variability and trait associations in bread wheat under drought stress, aiming to identify promising genotypes and key selection criteria for breeding drought-resilient varieties. The study employs a comprehensive approach, combining physiological, biochemical, and agronomic assessments with multivariate statistical analyses. While the manuscript presents valuable data and insights, several areas could be improved for clarity, rigor, and impact.

Strengths:

Comprehensive approach: The study evaluates a wide range of physiological, biochemical, and agronomic traits, providing a holistic view of drought responses in wheat.
Use of multivariate analyses: The application of PCA, cluster analysis, and AMMI analysis enhances the robustness of genotype classification and trait association studies.
Identification of promising genotypes: The study identifies several genotypes (G3, G8, G12) as promising candidates for drought tolerance breeding.
Emphasis on key selection criteria: The manuscript highlights the importance of chlorophyll content, RWC, MSI, photosynthetic efficiency, and antioxidant enzyme activities as potential selection criteria.
Weaknesses and Suggestions for Improvement:

While the abstract summarizes the main findings, it could be more concise and impactful. Focus on the most important findings and quantify the improvements observed in tolerant genotypes (e.g., yield increase under drought). Avoid excessive detail (e.g., listing all enzymes).

The introduction is lengthy and could be more focused. Streamline the background information on wheat production and the impact of drought.
The justification for using multivariate analyses could be strengthened. Clearly articulate why these methods are superior to traditional univariate approaches in this context.
The specific objectives of the study should be stated more clearly and concisely at the end of the introduction. What gap in knowledge is this research filling?

Materials and Methods:

Experimental Site: More details about the specific location (e.g., soil type classification, previous cropping history) and justification for its selection are needed. "Arid" is a general term; provide specific climatic data (e.g., average rainfall, temperature range, evapotranspiration) for the two growing seasons. The relevance of Figure S1 should be explicitly mentioned in the text.
Plant Materials: Provide more information about the selection criteria for the 14 genotypes. Why were these specific genotypes chosen? What is the basis for their "variability in yield and drought tolerance"? Table S2 should be integrated into the main text or moved to supplementary material. Clearly define what constitutes "advanced breeding lines," "exotic genotypes," and "high-yielding commercial cultivars."


Tables and Figures: Improve the quality and clarity of all tables and figures. Ensure that all labels are clear and concise. Add error bars to figures. Provide clear captions for all tables and figures.

Supplementary Material: Ensure that any supplementary material is clearly referenced in the main text.

Language and Grammar: The manuscript contains some grammatical errors and awkward phrasing. Careful proofreading is essential.

Overall:

This manuscript has the potential to make a valuable contribution to the field of drought tolerance breeding in wheat. However, significant revisions are needed to improve the clarity, organization, and rigor of the presentation. Addressing the weaknesses outlined above will significantly strengthen the manuscript and increase its impact.

Experimental design

Justify the choice of irrigation regimes. How were the drought stress conditions determined and validated? Were soil water content or other measures used to confirm the level of stress? The number of irrigations alone is not sufficient to define drought stress. Specify the timing of irrigations relative to wheat growth stages.
Measured Traits: Provide more details about the measurement protocols. For example, how were leaf samples collected for chlorophyll and RWC measurements? When were measurements taken (e.g., growth stage)? For photosynthetic parameters, explain how PAR was measured and integrated into the NPR calculation. Clarify the units for Tr (μmol CO2/m²/s is unusual; it's typically μmol H2O/m²/s). For MDA, specify the exact method used (e.g., TBA assay).
Statistical Analysis: Specify the software used for all statistical analyses. Justify the use of specific tests (e.g., Shapiro-Wilk, Bartlett's). Explain how drought tolerance indices were calculated (provide formulas). The rationale for using these specific indices should be stated. Clarify the criteria used for clustering genotypes.

Validity of the findings

The results section is very descriptive and lacks focus. Prioritize the most important findings and present them more concisely. Avoid simply listing all the data. Focus on trends and differences between genotypes and treatments.
Figures 2-4 are difficult to interpret due to the small size and lack of clear labels. Consider combining some figures or using a different format (e.g., box plots) to improve clarity. Add error bars to all figures.
The description of the heatmap and cluster analysis (Figure 5) is too detailed. Focus on the main clusters and their distinguishing characteristics. Relate these clusters back to the genotypes identified as promising in earlier sections.
The discussion of drought tolerance indices (Figure 6) should be more concise. Focus on the genotypes in the top-performing group and explain why they are considered drought-tolerant.
The descriptions of AMMI and PCA (Figures 7 and 8) are lengthy and difficult to follow. Focus on the key patterns and relationships. Clearly explain the interpretation of the biplots. What do PC1 and PC2 represent in terms of the measured traits?
The correlation and path analyses (Figure 9 and Table S5) should be presented more concisely. Focus on the most significant correlations and path coefficients. Explain the biological significance of these relationships.

·

Basic reporting

The manuscript entitled "Genetic variability and phenotypic associations of physiological and agronomic traits in bread wheat genotypes under drought-stressed and well-watered conditions" studied the behavior of a group of bread wheat genotypes under control and water stress conditions aiming to determine the genotypes performing better under drought conditions and to explore the relationships among different categories of traits. The results seem to be interesting and could be of practical use in breeding programs of bread wheat under Egyptian conditions. Nevertheless, the paper needs further edits and improvements in order to be more clear and with better impact.

The text should be thoroughly improved in terms of grammar, rephrasing and fluency like in lines 127 and 433.

Title
I suggest to make some edits on the title like:
Genetic variability and traits association for physiological and agronomic traits in bread wheat genotypes under drought-stressed and well-watered field conditions"

Introduction
The introduction overall sets the research in its context, however there is an obvious lack related to the Egyptian context, which should be provided in the introduction; aspects related to rainfall, irrigation and wheat productivity … and to set your objective based on these challenges.

Experimental site
The rainfall and temperature across the two cropping seasons should be described in a paragraph, and the comparison between the two season should be highlighted. If there is a significant difference in rainfall between the seasons (per month or during sensitive growth stage like flowering…), you should provide how did you handle that during the irrigation in the Experimental design and agronomic practices section.
The details of the nutrient status of the experimental site should be briefly described since that they are available in the Table S1.
Line 127: Replace “growth season” by “growing season” throughout the paper which is commonly used.
Line 129: use “filed capacity” instead of “field capacitance”

Plant materials
Provide the reference (research paper or other) displaying the variability of yield and drought tolerance of these wheat accessions (132-133)
Re-write this sentence and place it within the paragraph to be more fluent “The Egyptian Agricultural Research Center and the International Maize and Wheat Improvement Center (CIMMYT) provided these genotypes” (Lines 133-134).
The advance lines employed in this study belong to which generation, are they cultivated by the farmers?

Experimental design and agronomic practices
Lines 141-142: simplify the sentence to be more fluent, and provide the details about the timing of each irrigation; days after sowing and plant growth stage for each water treatment per growing season.
Mention the irrigation system used for watering.
Line 145: rephrase the following sentence “Six rows in each plot, each three meters long with 20-cm between rows”.
What about weeding? Add the timing and the active ingredient for weeding operations.
Trait measurement
The growth stage (timing) should be mentioned for all the traits you have assessed.
Line 197: replace “tons of ha-1” by “tons ha-1”
Remove the citation of Table S3 (line 208) in the statistical analyses section, this should be cited in the results part.

Results
Physio-biochemical attributes section (Line 231) was largely detailed; this section should be shortened.
Please carefully revise the section “Ranking and AMMI analyses” specifically the allocation of environments to negative or positive side of the PC1 and PC2. For instance, in the lines 333-334, the E2 and E4 are located on the positive side of PC1 not on the negative one. Also, in lines 339-340, E2 should be on the positive side of PC2 and on the negative side of PC1.

Discussion
Discuss how your results are useful for practical use, explaining the potential use of the tested genotypes (by their names not codes) and the assessed traits in wheat breeding programs for drought tolerance under Egyptian conditions.
Discuss the difference between the local and the exotic genotypes and if this has an effect on the use of your results in breeding programs.

Experimental design

'no comment'

Validity of the findings

'no comment'

---

## Round 0.2 · Minor Revisions

Your manuscript presents a valuable piece of research in a good way and it has gotten even better according to the reviewer's comments. Before its acceptance, however, you should correct the following points.

- Either use abbreviations throughout the text or do not. If you use an abbreviation, you should give it at the first occurrence of the word. Examine the entire manuscript carefully and explain the abbreviation the first time it appears in the text and then use the abbreviation.
- The description of the methods between lines 197-209 is complicated. Separate headings
- Explain more about Relative Water Content
- Give the formula for Membrane stability Index
- Explain MDA content separately. Correct that MDA indicates lipid peroxidation, not H2O2 level. MDA is incompletely explained. For detailed information on the methods, please refer to the article “Inter-subspecies diversity of maize to drought stress with physio-biochemical, enzymatic and molecular responses”.
- Briefly describe all the agronomic features.
- Line 255: delete the title. irrelevant
- Line 280: proline content will be fresh weight (FW), not dry weight (DW). This analysis is done on fresh leaf samples. Bates' method is so.
- You forgot to write the references in Table 1.

·

Basic reporting

I thank the authors for revising the manuscript taking into account the recommendation of the reviewers. I recommend the manuscript "Genetic variability and trait associations for physiological and agronomic characteristics in bread wheat genotypes under drought stress and well-watered conditions" to be accepted for publication in PeerJ journal.

Experimental design

'no comment'

Validity of the findings

'no comment'

Additional comments

'no comment'

---

## Round 0.3 · accepted · Accept

The changes you make are sufficient for the manuscript to be accepted. Congratulations.